# Bacterial Pericarditis Caused by Penetration of a Migrated Biliary Stent from the Lateral Segment of the Liver: A Case Report

**DOI:** 10.3390/medicina58010132

**Published:** 2022-01-15

**Authors:** Hsiao-Yun Chao, Chih-Huang Li, Shou-Yen Chen

**Affiliations:** 1Department of Emergency Medicine, Chang Gung Memorial Hospital and Chang Gung University, Linkou, Taoyuan City 333, Taiwan; b101092070@tmu.edu.tw (H.-Y.C.); chhli2002@gmail.com (C.-H.L.); 2Graduate Institute of Clinical Medical Sciences, Division of Medical Education, College of Medicine, Chang Gung University, Linkou, Taoyuan City 333, Taiwan

**Keywords:** biliary stent, stent migration, pericarditis, emergency department

## Abstract

Endoscopic biliary stent insertion is a well-established procedure that is indispensable in the management of various benign and malignant biliary disorders, and one that helps prevent mortality related to invasive surgical procedures. We report a rare case of the distal migration of a biliary stent outside the abdomen to the pericardium, inducing constrictive pericarditis and septic shock. This case alerts clinicians to be aware of potential adverse events that can lead to unfavorable patient outcomes. Such adverse events can be effectively avoided through early detection and intervention.

## 1. Introduction

Endoscopic placement of biliary endoprostheses is a well-established and minimally invasive procedure applied in patients with biliary stricture or obstruction [1]. Adverse events associated with stent use include occlusion due to clogging and stent migration. The typical sites of migration or perforation are the duodenum or the colon [2,3,4,5]. Migration of stents outside the gastrointestinal wall is less frequent but occasionally reported [6,7,8]. We present the case of a biliary stent that migrated from a lateral segment of the liver and penetrated the pericardium. The stent migration and penetration induced bacterial pericarditis one month after the stent replacement procedure. Distal pericardial involvement with a migrated biliary stent is unusual. Our objective in presenting this rare case is to share the information with the scientific community and to discuss an adverse event of biliary stent migration that has not, to our knowledge, been previously reported.

## 2. Case Report

An 80-year-old female patient underwent endoscopic retrograde biliary drainage because of choledocholithiasis-associated obstructive cholangitis in 2011. Dilation of common biliary and common hepatic ducts (up to 20 mm) with several radiolucent filling defects (up to 20 mm in diameter) was noted. Endoscopic sphincterotomy was performed and several pigmented stones were extracted. A double pigtail plastic stent (7 French, 9 cm) was inserted after complete clearance of bile ducts, which was demonstrated by cholangiogram. The patient developed symptoms of progressive chest tightness one month after the procedure and subsequently visited our emergency department for treatment. She reported experiencing chills but not fever. Abdominal pain, back pain, dyspnea, or orthopnea were also absent. A chest X-ray revealed marked cardiomegaly without active lung lesions, and an electrocardiogram revealed sinus tachycardia without significant ST segment change. Laboratory tests indicated significant leukocytosis (white blood cell: 27.5, 1000/µL), elevated C-reactive protein (210.52 mg/L), hyperbilirubinemia (total bilirubin: 3 mg/dL, direct bilirubin: 1.5 mg/dL), normal liver function (AST: 29 U/L), and a normal level of cardiac enzyme (troponin-I 0.04 ng/mL). An echocardiogram was performed and revealed that the pericardial space was occupied by a semisolid fluid, an indication consistent with constrictive pericarditis. Because of the unusual echocardiogram findings and presence of shock in the patient, a computed tomography (CT) scan of the chest was performed. The CT scan indicated a migrated plastic catheter from the left lobe of the liver penetrating the pericardium and resulting in pericardial effusion and changes consistent with pericarditis (Figure 1 and Figure 2). General and cardiovascular surgeons were consulted and surgical intervention was advised. The patient underwent an immediate operation in which the penetrating plastic biliary prosthesis was removed and 200 mL of serosanguinous pericardial effusion was aspirated. Lateral segmentectomy of the liver, cholecystectomy, choledocholithotomy with T-tube insertion, and pericardiectomy procedures were also performed. Cultures from the pericardial effusion revealed *Escherichia coli* and *Peptostreptococcus* species. The patient subsequently received parenteral antibiotics and was discharged after three weeks.

## 3. Discussion

The insertion of biliary endoprostheses has become a well-established and first-line nonsurgical procedure for diverse obstructive biliary and pancreatic disorders [9]. The procedure is generally safe and minimally invasive, although complications such as cholangitis, biliary stent occlusions, hemorrhage, duodenal perforation, pancreatitis, or biliary stent migration have been reported, with incidence ranging from 8% to 10% [5,10,11,12]. Biliary stent migration, either proximal or distal, is reported in 5% to 10% of patients undergoing biliary stent insertion [11,12,13]. The independent predictors for stent migration are moderate to marked common bile duct dilation, complete sphincterotomy, the use of balloon dilation, and the use of wide, straight stents inserted for more than one month [14]. In reported cases, biliary stent migration has been associated with complications such as penetration and intestinal obstruction or perforation [4,5,6,8,15,16,17,18]. To our knowledge, although these adverse events occur, no extra-abdominal penetration by biliary stents has been noted in earlier studies. The mechanism of penetration of the migrating plastic biliary stent through the diaphragm and into the pericardium remains uncertain. Old age, improper biliary stent size, necrotic changes in the lateral segment of the liver, and ischemia of the adjacent diaphragm tissue secondary to abdominal compartment syndrome may all result in increased intra-abdominal pressure and may contribute to this serious adverse event [19].

The most important element in preventing the adverse events of biliary endoprostheses is the appropriate use of the biliary stents according to the indications [20]. Endoscopic placement of a temporary biliary plastic stent is suggested in patients with irretrievable biliary stones [21]. Complete clearance of biliary stones was noted after endoscopic sphincterotomy in this patient, so placement of the biliary stent may not have been necessary and the adverse event could have been prevented from happening. In addition, the choice of appropriate length of stent is crucial, as improper stent length may increase the risk of duodenal injury. Distal migration was reported more likely to occur during the early period after biliary stenting. Close monitoring in the first three months after stenting for any unexplained abdominal discomfort is recommended [22,23]. Alternative strategies to mitigate the problem of plastic stent migration have also been reported with various success rates, including stent designs, such as self-expandable metal stents which were adapted to improve stent patency; refining the stent shape by mimicking the spindle shape of the common bile duct which narrows toward the papilla; the use of an anatomically-shaped stent instead of the typical cylindrical shape demonstrating a low migration rate through avoiding excessive radial expansible force inside the tube; or adding a double-pigtail plastic stent which anchors the fully-covered metal stent to prevent migration [24,25].

From the perspective of emergency care practice, although the stent had migrated outside the abdomen through the biliary tract, our patient did not exhibit obvious gastrointestinal symptoms such as abdominal pain, vomiting, jaundice, or dyspepsia. She reported chest discomfort instead of gastrointestinal symptoms; thus, physicians initially focused on etiologies of cardiovascular or respiratory origin. The combination of the presence of septic shock and the unusual pericardial effusion without cardiac tamponade prompted clinicians to explore other possible causes and arrange the CT scan that revealed the presence of the foreign body in the patient.

This case illustrates a rare adverse event of a biliary endoprosthesis, and it serves as a reminder to clinicians of the possibility of emergent adverse events related to implant insertion procedures. Because rapid and accurate diagnosis can be challenging in the absence of typical symptoms, stent migration–associated complications must be considered as a differential diagnosis whenever examining a patient with a recent history of an implant procedure.

## 4. Conclusions

Regular follow-up of implanted prostheses and devices such as biliary stents, especially in elderly patients, is crucial. In addition, distal migration events require additional attention because of the more severe adverse events that can occur as a result. Clinicians should be mindful of such circumstances to facilitate early discovery of any complications and to avoid poor patient outcomes.

## Figures and Tables

**Figure 1 medicina-58-00132-f001:**
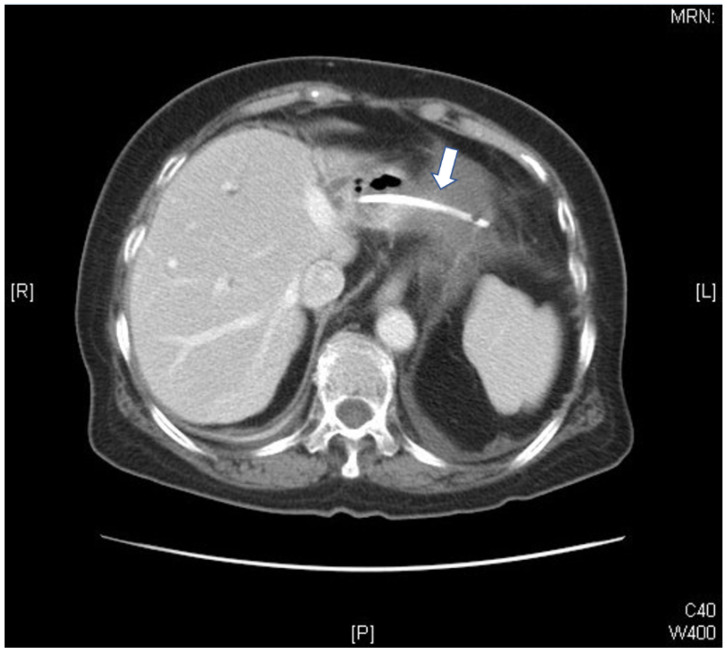
Migrated biliary stent in the abdomen.

**Figure 2 medicina-58-00132-f002:**
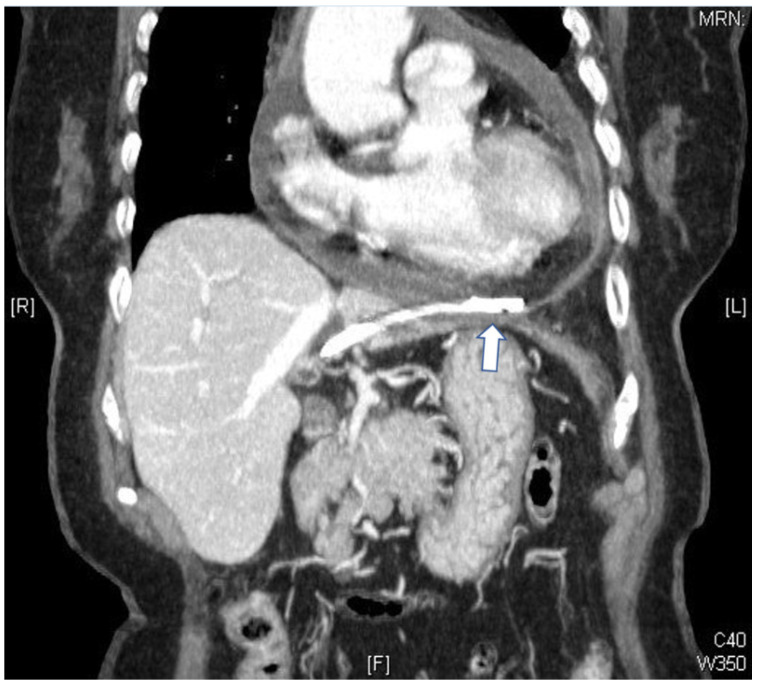
Migrated stent after penetration of the pericardium. One tip of the migrated stent was in the duodenum and the other tip penetrated into the pericardium through the lateral segment of left liver.

## Data Availability

The data that support the findings of this study are available from Linkou Chang Gung Memorial Hospital, but restrictions may apply to the availability of these data, which were approved by individual hospital IRB for the current study, and thus not publicly available. However, processed datasets can be requested and made available from the authors with the permission of Linkou Chang Gung Memorial Hospital.

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
