# Peer review of "Bacterial Pericarditis Caused by Penetration of a Migrated Biliary Stent from the Lateral Segment of the Liver: A Case Report"

_medicina, 2022, doi:10.3390/medicina58010132_

Round 1

Reviewer 1 Report

Thank you for giving me an opportunity to review this article.

This manuscript presents a rare adverse event after biliary stenting. This report is considered to be an interesting. However, I have few comments and questions regarding the manuscript.

Comments

#1

According to the ASGE guidelines on choledocholithiasis, long term biliary stenting may lead to stent-related adverse events (stent occlusion, migration, etc.). Therefore, it is recommended to remove stones as completely as possible, even in elderly patients. The authors should show why this patient was discharged with only a stent placement without stone removal in 2011.

#2

Stent migration may involve the characteristics of the stent and the anatomical factors of the bile duct. Please indicate the diameter, length, and shape of the stent. Also indicate the diameter of the bile duct and whether there was any bile duct stenosis. If possible, I recommend that you add pictures of the ERCP (the fluoroscopic and endoscopic images) as a Figure.

#3

Please indicate whether or not EST (endoscopic sphincterotomy) was performed during the ERCP.

#4

The discussion on metallic stents does not seem to have much relevance to this case report. It may be better to omit this part (Page3, Lines 81–88).

#5

As the authors argue, clinicians should be mindful of unanticipated adverse events in patients with biliary stenting. However, I believe that the most important is to prevent such adverse events from occurring. Please discuss how to prevent this sever adverse events before it occurs, at the viewpoint of the indication for stent placement, the shape of the stent, and other factors. In particular, the indication for stenting need more discussed. As I commented in #1, this adverse event might have been prevented if complete duct clearance had been performed at the initial ERCP.

#6

The authors use the term "complications" often. I suggest to replace it by the term "adverse events".

Author Response

#1

According to the ASGE guidelines on choledocholithiasis, long term biliary stenting may lead to stent-related adverse events (stent occlusion, migration, etc.). Therefore, it is recommended to remove stones as completely as possible, even in elderly patients. The authors should show why this patient was discharged with only a stent placement without stone removal in 2011.

ANS: Thank you for your comment. Actually, the stones were removed during ERCP. We are sorry that we did not describe it clearly. We have added this part into our text. Please see Line 35-39.

#2

Stent migration may involve the characteristics of the stent and the anatomical factors of the bile duct. Please indicate the diameter, length, and shape of the stent. Also indicate the diameter of the bile duct and whether there was any bile duct stenosis. If possible, I recommend that you add pictures of the ERCP (the fluoroscopic and endoscopic images) as a Figure.

ANS: Thank you for your comment. We have added this part into our text. Please see Line 35-39. However, the ERCP was done in other hospital and figures of the ERCP were not available. We hope you could understand. Thank you!

#3

Please indicate whether or not EST (endoscopic sphincterotomy) was performed during the ERCP.

ANS: Thank you for your comment. The EST was done. We have added this part into our text. Please see Line 35-39.

#4

The discussion on metallic stents does not seem to have much relevance to this case report. It may be better to omit this part (Page3, Lines 81–88).

ANS: Thank you for your suggestion. We have removed this part.

#5

As the authors argue, clinicians should be mindful of unanticipated adverse events in patients with biliary stenting. However, I believe that the most important is to prevent such adverse events from occurring. Please discuss how to prevent this sever adverse events before it occurs, at the viewpoint of the indication for stent placement, the shape of the stent, and other factors. In particular, the indication for stenting need more discussed. As I commented in #1, this adverse event might have been prevented if complete duct clearance had been performed at the initial ERCP.

ANS: Thank you for your suggestion. We have added this part into out text. Please see Line 87-103.

#6

The authors use the term "complications" often. I suggest to replace it by the term "adverse events"

ANS: Thank you for your suggestion. We have revised the term in most of the text according to your suggestion.

Reviewer 2 Report

Dear authors,

This is a very nice case report and very unusual. I have some minor comments embeded in the attached document. Please, adjust accordingly. 

Author Response

  1. No mention of stent replacement in case report

ANS: Thank you for your comment. We have added this part into our text. Please see Line 35-39.

  1. Describe the stent, manufacturer, size

ANS: Thank you for your comment. We have added this part into our text. Please see Line 35-39.

  1. Remove the date

ANS: Thank you for your suggestion. We have removed the date according to the suggestion.

  1. Is there any intra-operative image, image of the retrieved stent. Can you please describe in more detail the entry point on the pericardium. The stent was just protruding into the pericardium or there was lesion to some heart structures. Where was the other end of the stent, in the biliary tract, peritoneal cavity?

ANS: Thank you for your suggestion. We have added the details of the migrated stent in the figure legend of Figure 2. However, the intra-operative image is not available because the operation was done long time ago. We are sorry about this and we hope you could understand.

Round 2

Reviewer 1 Report

The manuscript has been much improved and is in a nice condition now.